# Small RNA Profiling of piRNAs in Colorectal Cancer Identifies Consistent Overexpression of *piR-24000* That Correlates Clinically with an Aggressive Disease Phenotype

**DOI:** 10.3390/cancers12010188

**Published:** 2020-01-12

**Authors:** Deepak Narayanan Iyer, Timothy Ming-Hun Wan, Johnny Hon-Wai Man, Ryan Wai-Yan Sin, Xue Li, Oswens Siu-Hung Lo, Dominic Chi-Chung Foo, Roberta Wen-Chi Pang, Wai-Lun Law, Lui Ng

**Affiliations:** Department of Surgery, Li Ka Shing Faculty of Medicine, The University of Hong Kong, Pokfulam, Hong Kong, China; deeiyer@hku.hk (D.N.I.); tmhwan@hku.hk (T.M.-H.W.); wysin17@hku.hk (R.W.-Y.S.); daisyxue@hku.hk (X.L.); oswens@hku.hk (O.S.-H.L.); ccfoo@hku.hk (D.C.-C.F.); lawwl@hku.hk (W.-L.L.)

**Keywords:** piRNA, colorectal cancer, prognosis, *piR-24000*, non-coding RNA, biomarker

## Abstract

Piwi-interacting RNAs (piRNAs) represent a novel class of small non-coding RNAs (ncRNAs) that have been shown to have a deregulated expression in several cancers, although their clinical significance in colorectal cancer (CRC) remains unclear. With an aim of delineating the piRNA distribution in CRC, we conducted a systematic discovery and validation of piRNAs within two clinical cohorts. In the discovery phase, we profiled tumor and adjacent normal tissues from 18 CRC patients by deep sequencing and identified a global piRNA downregulation in CRC. Moreover, we identified *piR-24000* as an unexplored piRNA that was significantly overexpressed in CRC. Using qPCR, we validated the overexpression of *piR-24000* in 87 CRC patients. Additionally, we identified a significant association between a high expression of *piR-24000* and an aggressive CRC phenotype including poor differentiation, presence of distant metastases, and a higher stage. Lastly, ROC analysis demonstrated a strong diagnostic power of *piR-24000* in discriminating CRC patients from normal subjects. Taken together, this study provides one of the earliest large-scale reports of the global distribution of piRNAs in CRC. In addition, *piR-24000* was identified as a likely oncogene in CRC that can serve as a biomarker or a therapeutic target.

## 1. Introduction

Colorectal cancer (CRC) is responsible for 10.2% of all new cancer cases, while being the second leading cause for cancer-related deaths worldwide [1]. Despite the fact that significant progress has been made in recent years to improve the strategies associated with early detection and treatment of the disease, the overall poor clinical outcome of advanced CRC patients remains a critical challenge [2,3]. Only via a better understanding of the molecular networks surrounding the growth, development, and spread of CRC can we identify effective therapeutic avenues to circumvent the issues associated with poor prognosis.

Like several solid tumors, CRC is a slow, multistep process that involves the interplay of several critical oncogenes and tumor suppressor genes [4,5,6,7]. Moreover, there is mounting evidence to indicate a crucial role of epigenetic alterations in the pathogenesis of CRC [8,9,10,11,12]. Specifically, the role of non-coding RNAs (ncRNAs) as critical epigenetic regulators responsible for the occurrence and progression of CRC has been recognized for some time now. Among the ncRNAs, microRNAs (miRNAs) have gained immense recognition in the past decade for their participation in a multitude of processes associated with the development of multiple types of cancer, including CRC [12,13,14,15]. Recently, another class of small ncRNAs known as P-element-induced wimpy (PIWI)-interacting RNAs (piRNAs) has garnered attention for their central roles in cellular homeostasis and regulatory roles in cancer and several other diseases [16]. PiRNAs are 26–31 nucleotide ncRNAs that were initially identified in mice testis for their germline role in preserving genomic stability by interacting with PIWI proteins [17,18,19,20]. In contrast to miRNAs, several hundred thousand piRNAs are transcribed from thousands of genomic loci, suggesting that the piRNA molecules are vastly diverse [21]. Through a complex biogenesis pathway, piRNA molecules exert their effect by repressing transposons as well as protein-coding genes or long ncRNAs, either by posttranscriptional regulation or other epigenetic mechanisms [20,22,23]. Recent studies have identified that the expression of piRNAs is frequently deregulated in a variety of cancers [24,25], although research in understanding the biological and clinical implications of piRNAs in CRC is still in its early stages. Nevertheless, results from piRNA-profiling studies in CRC have shown that dysregulated expression of a few piRNAs, including piR-823, piR-1245 and, piR-54265, is correlated with advanced tumor stage and overall poor clinical presentation of CRC [26,27,28]. Owing to their small size, high stability, and ease of detection, these piRNAs were thus classified as potent predictive biomarkers that could also serve as potential therapeutic targets in CRC. Nevertheless, the role of piRNAs in the growth, development, and spread of CRC remains largely elusive, and further investigations are required to interpret the molecular interplay of piRNAs in the pathogenesis of CRC.

With the advancement of high-throughput technologies such as next-generation sequencing, substantiated with improved data analysis platforms, large scale detection of novel molecular targets such as piRNAs has now become relatively simpler. However, there is highly limited literature available on large-scale piRNA profiling in CRC. Taking this limitation into consideration, we attempted to perform a systematic genomic profiling of piRNAs in CRC to understand the disease-specific global distribution of these molecules. Subsequently, we validated the expression of a novel piRNA in a larger independent patient dataset to identify its clinical correlation. Consequently, we were able to identify *piR-24000*, an unexplored, potentially oncogenic piRNA that is overexpressed in CRC and correlates with an aggressive phenotype.

## 2. Results

### 2.1. piRNAs Show a Global Repression in CRC

We performed a systematic discovery assessment of piRNA biomarkers using next-generation sequencing in 18 patients (17 CRC biopsies and 18 adjacent normal tissues; CRC tissue for one patient was not available). The median age for the discovery cohort was 59.5 years with an equal distribution of sexes. Most patients exhibited nodal and/or distant metastasis (Stage IV—16.7%, Stage III—77.8%), with primary tumors commonly present in the colon (61.1%) (Table 1). Overall, we detected 143 piRNAs that were differentially expressed (−2 ≥ fold-change ≥ 2; *p* < 0.05) in the malignant and the non-tumor tissues (Appendix A). Globally, we observed that piRNA expression was considerably lower in the tumor tissue compared with the adjacent normal tissue (32 upregulated piRNAs vs. 111 downregulated piRNAs in tumor tissue). Hierarchical clustering of piRNA expression profiles resulted in a clear segregation of the CRC tumor and non-malignant samples, apart from one tumor sample (Figure 1). This was an indicator of the strong involvement of piRNAs with the overall process of colorectal carcinogenesis, thereby supporting the notion that piRNAs have an important role in cancer biology [16,29].

Among the piRNA probes analyzed by small RNA sequencing, we identified *piR-24000* as an unexplored piRNA that was significantly (*p* = 0.0094) overexpressed in CRC tumors as compared to the adjacent normal tissue, with a log_2_-fold change of +3.16 (Figure 2). To further confirm our results for *piR-24000* expression in CRC, we performed a real-time PCR-based expression analysis in matched tissue biopsies obtained from 12 patients, comprising of primary CRC and adjacent normal tissue, as well as matched adenoma biopsies derived from the same patients. Expression of *piR-24000* was normalized with the expression of *RNU6B* and was represented as −∆Ct, to give a better illustration for the change in expression of the gene. The resulting data indicated that *piR-24000* was overexpressed in 75% (9/12) of CRC tissue biopsies as well as adenoma tissues, as compared to the non-malignant tissue. Specifically, over 5-fold increase of *piR-24000* was observed in the primary CRC tissue (median: −13.469 vs. −15.603, *p* = 0.0080) and the adenoma tissue (median: −13.374 vs. −15.603, *p* = 0.0258), as compared to the normal colonic mucosa. There was no statistically significant change in the expression of *piR-24000* between the adenoma and the CRC tissue groups (Figure 3A).

### 2.2. High Expression of piR-24000 Is Associated With Poor Clinical Presentation in CRC

In order to identify the association of *piR-24000* with patient clinicopathological parameters, we performed a validation of the previous data (next-generation sequencing and the pilot analysis in 12 patients) in a cohort of 87 patients using real-time PCR. The median age of the validation cohort was 70, comprising nearly 60% males. Of the 87 CRCs, 56 were colonic (64.4%), 23 (26.4%) were localized in the rectum, and the remaining (9.2%) had a rectosigmoid origin. Overall, 13 (14.9%) tumors were stage I, 20 (23.1%) were stage II, 33 (37.9%) were stage III, and the rest (24.1%) presented as stage IV tumor (Table 1).

Expression analysis of *piR-24000* in 87 pairs of CRC biopsies and their matched adjacent non-malignant tissues by real time PCR revealed that *piR-24000* was significantly overexpressed in 74.7% (65/87) (T/N > 2-fold) of CRC tissues. Overall, the CRC tissues exhibited a 6.2-fold increase in the expression of *piR-24000* compared to the normal tissue (median: −13.08 vs. −15.72, *p* < 0.0001, Figure 3B). Subsequently, the median relative expression level (−∆Ct) of *piR-24000* in primary CRC tissues (−13.08) was identified as the cut-off point to separate the tumors into median-high-*piR-24000*-expressing and median-low-*piR-24000*-expressing groups. Statistical analysis using Kruskal–Wallis test showed that the expression of *piR-24000* was positively correlated with tumor stage. While *piR-24000* was expressed at relatively lower or similar levels in stages I (8/13 cases with low *piR-24000*, median: −13.69), II (11/20 cases with low *piR-24000*, median: −13.54), and III (17/33 cases with low *piR-24000*, median: −13.30), there was a marked increase in the expression of *piR-24000* in stage IV patients (14/21 cases with high *piR-24000*, median: −12.57) (*p* = 0.0377) (Table 2).

Furthermore, presence of distant metastasis was also significantly associated with a higher expression of *piR-24000*, as compared to the patients showing no distant metastasis (median: −12.56 vs. −13.52, *p* = 0.0198) (Table 2). Liver (16/21 stage IV cases, 76.1%) was the primary site for distant metastasis, followed by lungs (3/21 stage IV cases, 14.3%). Consequently, we also profiled the expression of *piR-24000* in 20 distant liver metastases and found a strong overexpression of this piRNA in liver metastases as compared to non-metastatic CRC tissues (median: −11.79 vs. −13.53, *p* = 0.0023) as well as to the normal colonic mucosa (median: −11.79 vs. −15.72, *p* < 0.0001). No significant difference was observed in the expression of *piR-24000* between distant liver metastases and metastatic CRC tissue specimens (median: −11.79 vs. −12.57, *p* = 0.7276) (Figure 4).

The expression of *piR-24000* was also found to be remarkably higher in patients presenting with moderate (39/76 with high *piR-24000*, 51%, median: −13.03) and poor (4/6 cases with high *piR-24000*, 66.7%, median: −10.03) tumor differentiation, as compared to the CRC tumors with well-differentiated presentation (1/6 cases with high *piR-24000*, 16.7%, median: −15.61) (*p* = 0.0133) (Table 2).

Although a positive association was observed between the expression of *piR-24000* and advanced nodal metastasis (16/28 N2 cases showing high *piR-24000* expression) and with advanced tumor invasion (12/19 T4 cases showing high *piR-24000* expression), these association were not statistically significant (Table 2). None of the other clinicopathological parameters, including age, sex, tumor location, or tumor size, correlated significantly with the expression level of *piR-24000* (Table 2).

### 2.3. Evaluation of Tissue piR-24000 as a Potential Biomarker in CRC

We plotted receiver operating characteristic (ROC) curves to estimate the diagnostic accuracy of *piR-24000* in CRC. As shown in Figure 5, *piR-24000* could significantly differentiate CRC patients from normal subjects with an area under curve (AUC) value of 0.8175 (95% confidence interval = 0.7521 to 0.8830, sensitivity = 93.1%, specificity = 68.97%, *p* < 0.0001) (Figure 5A). Subsequently, we bifurcated the patients into early-stage (stages I and II) and late-stage (stages III and IV) groups and plotted individual *piR-24000*-based ROC curves for the groups. Within the late-stage group, we observed an AUC value of 0.8405 (95% confidence interval = 0.7610 to 0.9200, sensitivity = 96.3%, specificity = 70.37%, *p* < 0.0001), and in the early-stage group we obtained a slightly lower AUC value of 0.7796 (95% confidence interval = 0.6648 to 0.8944, sensitivity = 87.88%, specificity = 66.67%, *p* < 0.0001). Overall, these data suggest that *piR-24000* can strongly discriminate between CRC patients and control subjects.

## 3. Discussion

PiRNAs represent a large family of small non-coding RNAs that have recently been identified as potent gene modulators affecting the development and progression of multiple types of cancers. Within CRC, there are limited examples of piRNAs that have been identified and explored to date. Consequently, the present study was designed with the aim of exploring and characterizing the piRNA profile in CRC. Martinez et al. performed the earliest large-scale piRNA profiling efforts in normal and cancerous tissues from The Cancer Genome Atlas (TCGA) database, which panned across multiple organs, identifying a large-scale deregulation in the global levels of piRNAs, simultaneously revealing that piRNA expression patterns are distinct in tumors as compared to the adjacent non-malignant tissues [30]. Owing to insufficient non-malignant colorectal tissue-related data within the TCGA database, the authors could not provide a robust analysis for the piRNA distribution within CRC. Taking this into consideration, the present study was the first systematic piRNA-profiling effort in CRC that was able to identify a pool of deregulated piRNAs showing distinct expression patterns in malignant colorectal tissue as compared to the adjacent normal tissue. Furthermore, we observed that a larger pool of piRNAs are downregulated in CRC specimens, indicating that piRNAs may have a largely tumor-suppressive function in CRC. While piRNAs are commonly reported as being upregulated in several cancers [30], there is contrasting evidence of global repression of piRNAs in some cancers, such as renal cell carcinoma [31,32]. Moreover, there is a huge literature gap in our understanding of the clear roles of piRNAs in somatic cells, which adds to the confusion of the specific roles of these small ncRNAs in cancer. Nevertheless, the dysregulated expression of piRNAs shown in the current study between the malignant and the adjacent normal tissue is the first large-scale evidence of a potential contribution of piRNAs to the process of colorectal carcinogenesis.

We further studied the role of an unexplored piRNA, *piR-24000*, in a larger validation cohort. *PiR-24000* (Accession: DQ593752; Aliases: piR-33864, PIR54863, hsa_piR_017184) is a 28-nucleotide small ncRNA with a single genomic location on Chromosome 13 [19]. Studies in multiple cancers have shown a consistent upregulation of *piR-24000*, indicating a strong role of this piRNA in cancer. A genome-wide piRNA-profiling study on 104 breast cancer patients identified a 9.1-fold upregulated expression of *piR-24000* (piR_017184) in breast cancer tissue specimens compared to normal control breast tissues [33]. Martinez et al. attempted to identify the signatures of gastric cancer recurrence by analyzing the transcriptomes of 320 gastric cancer and 38 non-malignant stomach tissues and identified a 6.3-fold overexpression of *piR-24000* (FR326119) in gastric adenocarcinoma as compared to the non-cancerous stomach tissues [34]. Furthermore, embedded within the *Tumor Protein, Translationally-Controlled* (*TPT1*) gene, which is known to be oncogenic in several cancers [35,36,37], the small nucleolar RNA 31 (SNORA31-001)–*piR-24000* (piR_017184) pair was found to be significantly overexpressed in breast cancer tumor tissues as compared to normal breast tissues [38]. Additionally, within CRC there are two reports to date indicating a significantly upregulated expression of *piR-24000* in malignant colorectal tissues; however, no clinical correlation was provided for this piRNA by either of the studies [26,39]. In line with these previous reports, our study reported a significant overexpression of *piR-24000* within the CRC tissue specimens as compared to the adjacent normal tissues. Additionally, we also identified an increased expression of this piRNA within the matched adenoma samples, suggesting that *piR-24000* strongly contributes to the overall process of carcinogenesis. Within the validation cohort, we found that *piR-24000* correlates with poor tumor differentiation and advanced tumor stage, as well as the presence of distant metastasis. Indeed, a large increase in the expression of *piR-24000* was also observed with late nodal stage and with advanced tumor invasion, though the correlation was not significant. Taken together, this indicates that a higher expression of *piR-24000* correlates with an aggressive CRC phenotype, which was also reflected in the high expression of this piRNA in distant liver metastases. Furthermore, by plotting a ROC curve, we were able to demonstrate that *piR-24000* can significantly discriminate between CRC patients and control subjects. Although a relatively low specificity was observed for this piRNA within this study, the true negative detection rate could be increased by using *piR-24000* in combination with other similar piRNA/miRNA-based biomarkers.

The overall strength of this study is that it provided one of the earliest systematic evidence of the profile of piRNAs in CRC. Furthermore, we were able to provide strong clinical evidence suggesting that *piR-24000* is potentially oncogenic and may serve as a biomarker in CRC, specifically in patients presenting with an advanced, aggressive clinical phenotype. Consequently, to elucidate the molecular mechanisms influencing the upregulation of *piR-24000*, there is a requirement of functional studies to investigate the mechanistic role, potential targets, and the overall molecular interplay exhibited by this piRNA in CRC.

## 4. Materials and Methods

### 4.1. Patients and Specimens

Fresh tumor and adjacent normal tissue specimens were obtained from CRC patients who underwent surgical resection at the Department of Surgery, Queen Mary Hospital, University of Hong Kong. Immediately after collection, the tissue samples were snap-frozen in liquid nitrogen and stored at −80 °C until further use. Clinicopathological information for the patients was obtained from the clinical management system of the hospital. The study was approved by the Institutional Review Board, University of Hong Kong (Ethical approval number: UW 17-416; Date of approval: 01-Nov-2017), and informed written consent was obtained from all study participants. Other tissue specimens (adenoma, liver metastases) used in this study were obtained similarly.

### 4.2. RNA Extraction

Total RNA was extracted from the flash-frozen tissue specimens using the mirVana miRNA isolation kit (Thermofisher, Massachusetts, United States) according to the manufacturer’s guidelines.

### 4.3. Small RNA Sequencing and Data Analysis

As a part of the discovery dataset, RNA extracted from 18 patients (17 tumor and 18 adjacent non-malignant tissues; 1 tumor specimen was unavailable) was used for small RNA sequencing. Small RNA library preparation and next-generation sequencing was carried out by Beijing Genomics Institute (BGI, Shenzhen, China). Briefly, the quality and quantity of RNA extracted was assessed by a bioanalyzer (Agilent, California, United States) and a nanodrop (Thermofisher, Massachusetts, United States). Only RNA samples displaying a 28S/18S ratio > 2 and an RNA integrity number > 7 were considered of a high quality and were used for small RNA library preparation using a Truseq Small RNA library preparation kit (Illumina, California, United States) as per the manufacturer’s guidelines. Later, the libraries were used for 50 bp single-ended small RNA sequencing on the HiSeq2000 (Illumina, California, United States) platform. Data analysis of the reads was carried out using PartekFlow^TM^ (Partek Inc., Missouri, United States). Briefly, for the raw data, adapter sequences (TGGAATTCTCGGGTGCCAAGG) were trimmed from both ends with a minimum read length of 20 bases. Subsequently, a quality read trimming was carried out to filter reads showing a minimum quality level > 20. The filtered clean reads were then aligned to the human genome (hg19 reference index) using Bowtie-1.0.0 [40], with a seed length of 15 and a seed mismatch limit of 1. The resultant .bam files were quantified to the hg19 piRNA transcript model obtained from piRBase (www.piRBase.org) [41,42]. Differentially expressed piRNAs between the CRC and adjacent non-malignant samples were determined using DESeq2-3.5 [43], using the Wald hypothesis test. PiRNAs showing a fold change of ≤ −2 or ≥2 and a *p*-value ≤ 0.05 were considered significant and reported.

### 4.4. cDNA Synthesis and Quantitative Real-Time Polymerase Chain Reaction

Expression of *piR-24000* within the tissue specimens was analyzed by reverse transcription followed by SYBR-green-chemistry-based quantitative PCR. Universal oligo dT primer (for reverse transcription) and amplification primers (for quantitative PCR) were designed as per Balcells et al. [44]. For the reverse transcription reaction, 100 ng of total RNA was polyadenylated using *E. coli* Poly(A) Polymerase (New England Biolabs, Massachusetts, United States), and was subsequently reverse transcribed using the PrimeScript RT Reagent Kit (Takara, Kyoto, Japan) as per the manufacturer’s guidelines. Quantitative estimation of *piR-24000* was subsequently carried out by the ABI VIIa PCR system (Applied Biosystems, California, United States) using the TB Green Premix Ex Taq II (Takara, Kyoto, Japan) kit. All experiments were performed in duplicate and results were normalized to the expression of *RNU6B* and expressed as −∆Ct (negative delta Ct). Primer sequences used are listed in Appendix A.

### 4.5. Statistical Analysis

All statistical analyses were conducted using GraphPad Prism 8.0.1 (GraphPad, California, United States). Normal distribution of each dataset was tested via Shapiro–Wilk test prior to analysis. A two-tailed sample Student’s *t*-test was used for analyzing paired sample groups. For analyzing the relationship of *piR-24000* with the clinicopathological characteristics, the Mann–Whitney test or Student’s *t*-test used for comparison between two groups. For three or more groups, one-way ANOVA or Kruskal–Wallis test was applied. *p*-values < 0.05 were considered statistically significant.

## 5. Conclusions

Our study provides the first report that piRNAs are strongly dysregulated in CRC as compared to the adjacent non-malignant tissues. Furthermore, we provided systematic evidence to show that *piR-24000* is frequently upregulated in CRC, specifically in patients presenting with an aggressive phenotype including poor tumor differentiation, advanced stage, and presence of distant metastasis. Lastly, by plotting ROC curves, we were able to demonstrate the diagnostic power of *piR-24000* in discriminating CRC patients from normal subjects. These results indicate that *piR-24000* is potentially oncogenic in CRC and can also serve as a novel biomarker. Further understanding of the precise roles played by *piR-24000* by means of functional investigations will not only improve our understanding of the biology of this piRNA in CRC, but will also provide direction towards translating the value of *piR-24000* to clinics as a biomarker as well as a potential therapeutic target.

## Figures and Tables

**Figure 1 cancers-12-00188-f001:**
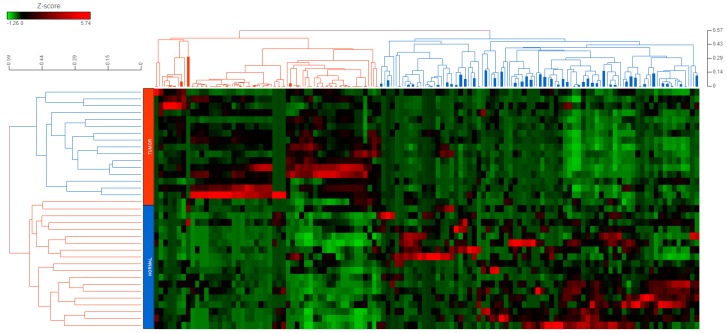
Hierarchical clustering of differentially expressed piRNAs in colorectal cancer. A total of 143 differentially expressed piRNAs (−2 ≥ fold-change ≥ 2; *p* < 0.05) in colorectal cancer (CRC) vs. non-malignant colon tissue were hierarchically clustered together using an average-linkage-based cluster distance metric and Pearson correlation as the point distance metric. Rows represent tissue specimens (Blue—normal tissue; Red—tumor tissue), while the columns represent the differentially expressed piRNAs. As per the Z-score color coding, red represents a high expression, whereas green represents a low expression of a given piRNA.

**Figure 2 cancers-12-00188-f002:**
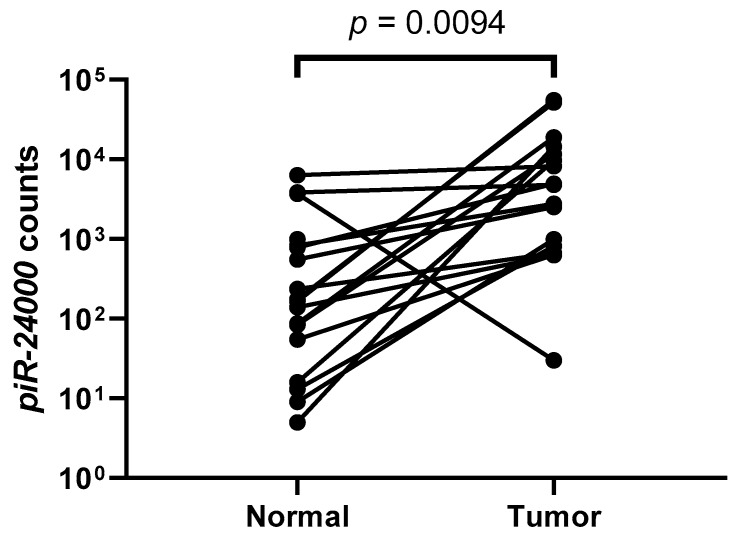
Expression of *piR-24000* in the discovery cohort. Expression levels (counts) of *piR-24000* in tumor samples compared with the adjacent non-malignant colon tissue.

**Figure 3 cancers-12-00188-f003:**
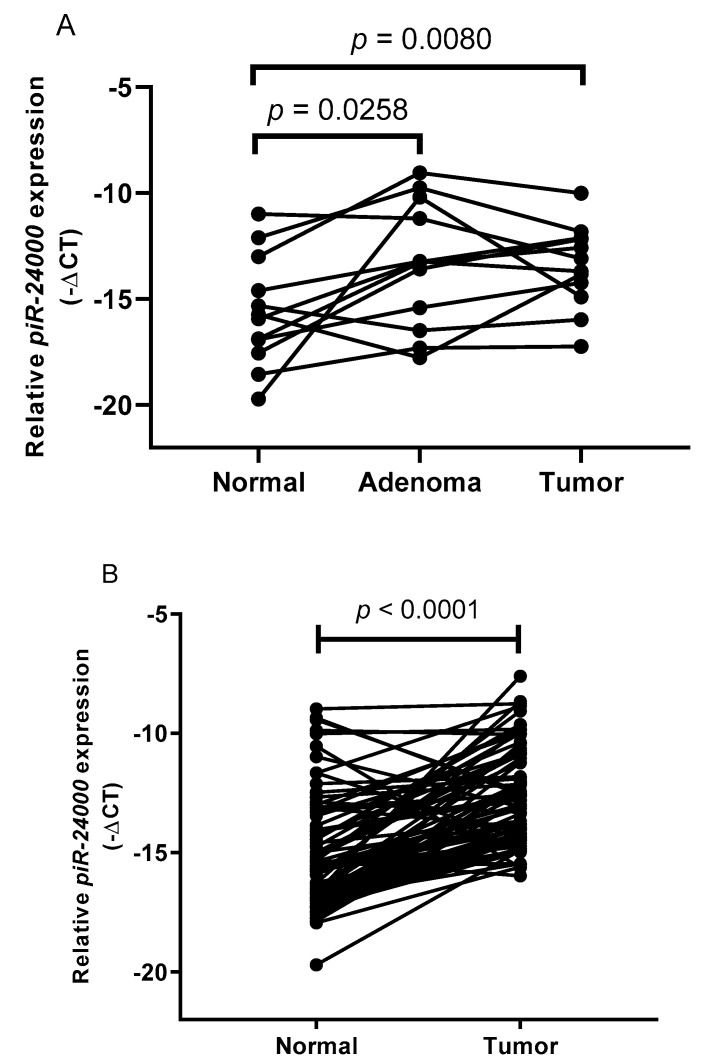
*piR-24000* is overexpressed in CRC. Real-time PCR-based expression analysis of *piR-24000* in (**A**) matched CRC, adenoma, and normal colon tissues from 12 patients, (**B**) matched CRC and adjacent normal colonic mucosa from 87 patients. Expression of *piR-24000* was normalized to *RNU6B* and expressed as −∆Ct (negative delta Ct).

**Figure 4 cancers-12-00188-f004:**
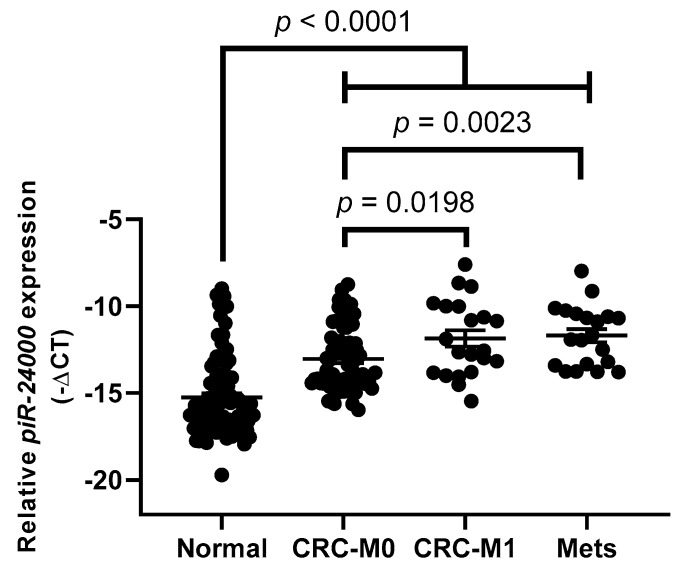
Expression of *piR-24000* in normal colonic mucosa and metastatic subtypes of CRC. Real-time PCR-based expression analysis of *piR-24000* in normal colonic tissue (*N* = 87), non-metastatic CRC tissue (CRC-M0; *N* = 66), metastatic CRC tissue (CRC-M1; *N* = 21), and distant liver metastases (Mets; *N* = 20). Expression of *piR-24000* was normalized to *RNU6B* and expressed as −∆Ct (negative delta Ct).

**Figure 5 cancers-12-00188-f005:**
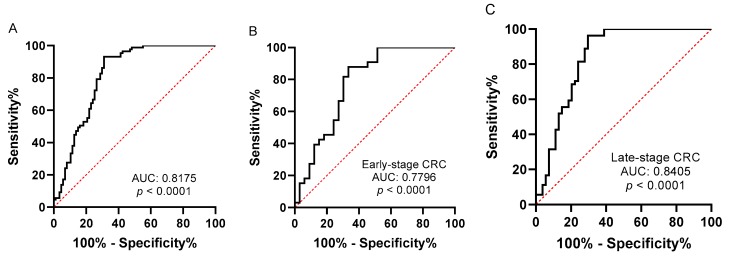
Assessment of the diagnostic performance of *piR-24000* in CRC. A standard receiver operating characteristic (ROC) curve was plotted for *piR-24000* to interpret its accuracy in discriminating CRC patients and control subjects in the following groups: (**A**) all patients combined, (**B**) early-stage CRC patients (stage I and II, *n* = 33), and (**C**) late-stage CRC patients (stage III and IV, *n* = 54).

**Table 1 cancers-12-00188-t001:** Characteristics of patients in the discovery and validation cohorts.

Characteristics	Category	Discovery Cohort	Validation Cohort
Age (median (range))		59.5 (40–79) years	70 (29–97) years
Gender (numbers (%))	Male	9 (50%)	51 (58.6%)
	Female	Female, 9 (50%)	36 (41.4%)
Tumor Location (numbers (%))	Colon	11 (61.1%)	56 (64.4%)
	Rectosigmoid	2 (11.1%)	8 (9.2%)
	Rectum	5 (27.8%)	23 (26.4%)
Tumor size (median (range))		40 (10–80) mm	40 (10–130) mm
Tumor differentiation (numbers (%))	Well-differentiated	0 (0%)	5 (5.7%)
	Moderate	16 (88.9%)	76 (87.4%)
	Poor	2 (11.1%)	6 (6.9%)
TNM stage (numbers (%))	I	0 (0%)	13 (14.9%)
	II	1 (5.5%)	20 (23.1%)
	III	14 (77.8%)	33 (37.9%)
	IV	3 (16.7%)	21 (24.1%)

Abbreviations: TNM (Tumor Node Metastasis).

**Table 2 cancers-12-00188-t002:** Correlation between clinicopathological features and *piR-24000* expression in 87 colorectal cancer tissues.

Characteristics	Category	*piR-24000* Expression (Numbers)		*p*-Value
		Median-Low	Median-High	
Age (Years)	<66	15	21	0.2928
	≥66	28	23	
Sex	Male	22	29	0.1471
	Female	21	15	
Tumor Location	Colon	26	30	0.5877
	Rectum (including rectosigmoid)	17	14	
Tumor Size	<45 mm	23	23	0.3236
	≥45 mm	20	21	
Tumor Differentiation	Well-differentiated	5	1	0.0133 *
	Moderate	37	39	
	Poor	2	4	
T Classification	T1–T2	10	7	0.4797
	T3	26	25	
	T4	7	12	
Lymph Node Metastasis	N0	20	18	0.4539
	N1	11	10	
	N2	12	16	
Distant Metastasis	No	36	30	0.0198 *
	Yes	7	14	
TNM Stage	I	8	5	0.0377 *
	II	11	9	
	III	17	16	
	IV	7	14	

* *p* < 0.05. Abbreviations: T classification (Tumor classification), TNM (Tumor Node Metastasis).

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
