# Peer review of "Small RNA Profiling of piRNAs in Colorectal Cancer Identifies Consistent Overexpression of *piR-24000* That Correlates Clinically with an Aggressive Disease Phenotype"

_cancers, 2020, doi:10.3390/cancers12010188_

Round 1

Reviewer 1 Report

In this study the authors performed piRNA profiling of CRC tissue vs adjacent normal tissue using NGS in  18 patients and a targeted study on piR24000 using real-time PCR in 87 CRC patients. They found that piR24000 was highly expressed in cancer tissue and propose that it may be an oncogene and a biomarker or therapeutic target. This is an interesting study and well written. 

Some comments:

In the list of overexpressed piRNAs in the appendix, piR24000 was not the top hit, neither was it anywhere close to the top. Presumably this piRNA was chosen for validation because of previous studies on its role in colorectal and other cancers as mentioned in the discussion. Why this piRNA was chosen for further study and not the others needs to be explained. Were the top hits e.g., piR2107 and 28160 able to be validated by real-time PCR?

RNU6B was used for normalization. The use of RNU6B for normalization has been questioned e.g., Schwarzenbach et al Clin Chem 2015 PMID 26408530. Was any other method used for normalization e.g., normalizing to the global average piRNA expression, and were the same results obtained?

The authors claim that pi24000 may be a good biomarker. There are no ROC-AUC studies to look at the performance of pi24000 to predict colorectal cancer. They used the median expression in CRC tissue to divide the expression into high and low expressors. An ROC curve would be better to derive a cut-off.

The heat map shows 2 different clusters for normal tissue and 3 different clusters for CRC tissue. Was there any difference in the groups e.g. location of tissue or stage of differentiation of CRC?

Are all the patients treatment-naive or did some receive neoadjuvant treatment before the surgery? Treatment may alter the piRNA expression in the tissue.

Reviewer 2 Report

The article by Iyer et al. describes the study of dysregulation of piRNA in colorectal cancer (CRC). The authors have compared the expression profile of cancer and normal tissue-derived RNA to examine the piRNA profile. The study is interesting as it explores a new species of non-coding RNA in cancer pathology, however the study, in my opinion, is not complete and is non-conclusive. I believe authors should include the following aspects in the study:

What is the clinical outcome of piRNA-24000 in disease progression. What is the mechanistic role of piRNA-24000 and other piRNAs in CRC pathobiology. What information is available regarding the treatment regimen given to each patient and what is the impact of treatment on piRNA profile? How piRNA profile can be used as diagnostic biomarker when most of the tissues are obtained from patients in advance stages of disease. Can authors make a comparison between cancer patients and normal individuals.  What authors mean by normal tissues can they elaborate exact location from they obtained the tissues and also how much tissues were used for RNA isolation.

Reviewer 3 Report

This study is describing the role of piR-24000 in colorectal cancer. Authors profiled both healthy and tumor tissue from 18 colorectal patients, and found that this piRNA is highly overexpressed in tumor tissue which could serve as a biomarker. The manuscript is interesting, the methods are chosen properly, however there are major concerns which needs explanation.

Major concern:

Authors describe piR-24000 as a novel piRNA, however, it is not newly described piRNA. Maybe they meant novel connection of this miRNA concerning colorectal cancer. It should be explained. The methodology is not sufficiently described. It is extremely difficult to find out how RNAseq data were analyzed, what were the cut-offs of the analysis. This section needs significant improvements. Both Tables presented in the manuscript are difficult to understand. What does the numbers presented in the tables correspond to. It’s probably the number of samples however it should be stated. Moreover, Table 2 show statistics corresponding to low and high expression of piR-24000, however no explanation of what high and low corresponds to was stated. 3 and 4 presents data of piR-24000 level of expression. There are 2 common ways to visualize such data: absolute analysis of gene expression based on standard used and relative analysis of gene expression based on internal control like housekeeping gene/miRNA. Here, authors used the second one, however it should be presented as ΔΔCT or 2- ΔΔCT, not the –ΔCT. This should be corrected as well as the description in the text concerning piR-24000 expression data. English needs editing.

Round 2

Reviewer 2 Report

I believe that the manuscript is in good shape now and authors have addressed all the queries of reviewers satisfactorily and the manuscript can now be accepted for publication.

Reviewer 3 Report

I am satisfied with the amendments made in the text. Manuscript can be accepted in present form.